# Analyzing Safe Haven, Hedging and Diversifier Characteristics of Heterogeneous Cryptocurrencies against G7 and BRICS Market Indexes

**Manoel Fernando Alonso Gadi** *,† 🆔 **and Miguel-Angel Sicilia** † 🆔

Department of Computer Science, University of Alcala de Henares, 28805 Alcala de Henares, Spain
* Correspondence: manoel.gadi@uah.es
† These authors contributed equally to this work.

**Abstract:** Cryptocurrency markets have experienced large growth in recent years, with an increase in the number and diversity of traded assets. Previous work has addressed the economic properties of Bitcoin with regards to its hedging or diversification properties. However, the surge of many alternatives, applications, and decentralized finance services on a variety of blockchain networks requires a re-examination of those properties, including indexes from outside the big economies and the inclusion of a variety of cryptocurrencies. In this paper, we report the results of studying the most representative cryptocurrency of each consensus mechanism by trading volume, forming a list of twenty-four cryptocurrencies from the 1st of January 2018 to the 30th of September 2022. Using the Baur and McDermott model, we examine hedge, safe haven, and diversifier properties of all assets for all G7 country's major indexes as well as all BRICS major indexes breaking it down by two attributes: kind of blockchain technology and pre/during COVID health crisis. Results show that both attributes play an important role in the hedge, safe haven, and diversifier properties associated with the asset. Concretely: stablecoins appear to be the only ones to maintain hedge property in most analyzed markets pre- and during-COVID; Bitcoin investment properties shifted after the COVID crisis started; China and Russia stopped being correlated with the cryptocurrency after the COVID crisis hit.

**Keywords:** diversifier; safe haven; hedging; G7; BRICS; blockchain technology; blockchain privacy; cryptocurrency

## 1. Introduction

Bitcoin is the largest and first widely adopted cryptocurrency (Demir et al. 2018). It is a decentralized system existing only in digital form–as a ledger any user can access—and is not governed or issued by any single government or banking entity (Nakamoto and Bitcoin 2008). Given its history of volatility, it is regarded as a high-risk investment. Despite this, due to its innovative and transparent qualities, Bitcoin has continued to grow in popularity since its inception in 2009. Most literature focuses on the safety and legality of Bitcoin, but in recent years an increasing number of articles address the hedge, safe haven, and diversifier properties of cryptocurrencies (Shahzad et al. 2020; Bouri et al. 2017; Dyhrberg 2016; Kang et al. 2020; Beneki et al. 2019; Wang et al. 2019; Jin et al. 2019; Chan et al. 2019; Garcia-Jorcano and Benito 2020; Mensi et al. 2019). Further, the market landscape has changed in recent years, with a variety of new programmable blockchain networks offering alternatives to Bitcoin. Some of them are built with special features (e.g., privacy in the case of ZCash) or platforms targeted to smart contracts that extend their base functionality (e.g., Ethereum). Thus, they might influence how their market behaves beyond pure financial concerns.

The most important added value of this work in comparison to the existing literature is the inclusion indexes of developing countries, as developing countries are leaders in

trading cryptocurrency asset and the interesting results achieved for those indexes makes this research an interesting tool for other researchers and investors.

Additionally, we compare and contrast other cryptocurrencies using the same methodologies to explore how they correlate or deviate from Bitcoin and market indices like the G7 and BRICS markets and how properties changed during pre-COVID and During-COVID.

The remainder of the paper is as follows. Section 2 describes previous relevant work. Section 3 describes our data, model, and methodology. Section 4 reports the results and discusses the findings. Then, Section 5 provides conclusions and an outlook.

## 2. Background

Previous work has reported research on the investment properties of Bitcoin, concretely the properties of Diversifier, Hegde, and Safe Haven. In what follows, we survey the main results to date.

Fang et al. (2019) provide evidence supporting that global economic policy uncertainty affects the long-run volatilities of Bitcoin and some other global equities/commodities but does not affect bonds.

Bouri et al. (2017) used a dynamic conditional correlation model (DCC) to examine whether Bitcoin could act as a cover and refuge for the main world stock indexes, oil bonuses, gold, raw materials, and the US dollar index. The results show that Bitcoin could not be considered a weak haven for the studied assets. The results of the DCC model show that Bitcoin can serve as an effective diversifier in most cases, and in a few cases, Bitcoin has been shown to have safe haven properties. Dyhrberg (2016) uses asymmetric GARCH methodology to show that bitcoin can clearly be used as a hedge against stocks in the Financial Times Stock Exchange Index. Kang et al. (2020) used a variation of the DECO-FIGARCH model and found a high negative correlation during 2011–2013 (European sovereign debt crisis), which demonstrated that Bitcoin could be a hedge asset during market turmoil and that Bitcoin reduces downside risk and reinforces the diversification advantages of equity portfolios during down trending market (bear market). It concludes that Bitcoin is a safe haven ONLY for the S&P 500. In the same direction, Shahzad et al. (2020) compares gold and Bitcoin to decide which of the two is a better safe haven, hedge, or diversifier asset for financial markets of the G7 countries (Canada, France, Germany, Italy, Japan, the UK, and the US), it concludes that gold is a safe haven and hedge for the G7. Bitcoin only shows safe haven and hedge properties for Canada. It concludes that the hedging effectiveness of gold is much higher than that of Bitcoin (Except in Canada).

Beneki et al. (2019) investigated how the emergence of Ethereum affects hedging in Bitcoin. This article is especially interesting because it is one of the first to measure volatility spillovers using the BEKK-GARCH model between a mature cryptocurrency (Bitcoin) with a less mature one (Ethereum). Wang et al. (2019) extended the work of Beneki et al. (2019) using a VAR-GARCH-BEKK model centering its attention on China, a country outside developed markets with much interest in Digital Currency Kharpal (2021).

Jin et al. (2019) included crude oil in the list of hedging assets with gold and Bitcoin. It concludes that the dynamic correlations between gold and crude oil markets are almost positive, indicating that crude oil is a diversifier for gold. However, dynamic correlations between Bitcoin and oil markets are nearly negative, indicating hedge properties.

Chan et al. (2019) expanded the list of market indexes to be Euro STOXX, Nikkei, Shanghai A-Share, S&P 500, and the TSX Index and used Pairwise GARCH models and constant conditional correlation with daily, weekly, and monthly returns from October 2010 to October 2017, concluding that Bitcoin is an effective strong hedge for all these indices under monthly data frequency. Garcia-Jorcano and Benito (2020) looked into similar markets (S&P500 (US), STOXX50 (EU), NIKKEI (Japan), CSI300 (Shanghai), and HSI (Hong Kong)) and confirmed that Bitcoin act as a hedge asset against the stock price movements of all international markets using several copula models, with Gaussian and Student-t copulas being the most relevant.

Mensi et al. (2019) created three types of mixed portfolios (risk-minimizing portfolio, equally weighted portfolio, and hedging portfolio) formed of Bitcoin, Dash, Ethereum, Litecoin, Monero, and Ripple, and showed evidence that a mixed portfolio provides better diversification benefits for investors and portfolio managers.

After reviewing the available literature, we identified the need for further research in two main directions. First, including more reference market indexes outside the big economies, is especially important because owners of cryptocurrencies globally come from outside big economies Sta (2021). Secondly, including additional cryptocurrencies in order to explore investment patterns through Blockchain Technology and maturity (market adoption). On top of that, Table 1 shows a series of articles applying Baur and McDermott's (2010) model for identifying Haven and Hedge properties to one or more asset(s). Therefore, the methodology chosen is Baur and McDermott (2010), which according to Table 1, continues to appear often over the years.

**Table 1.** Articles over the years that applied the Baur and McDermott (2010) model.

| Year of Publication | Reference | Asset Analyzed | Method Used |
|---|---|---|---|
| 2010 | Baur and McDermott (2010) | Gold | Baur and McDermott (2010) |
| 2016 | Baurand and McDermott (2016) | Gold | Baur and McDermott (2010) |
| 2017 | Bouri et al. (2017) | Major world stock indices, bonds, oil, gold | Baur and McDermott (2010) and Baur and Lucey (2010) |
| 2019 | Sett (2019) | G7 Stock Indexes and Bonds | Baur and McDermott (2010) and Baur and Lucey (2010) |
| 2020 | Baur and Kuck (2020) | S&P 500 | Baur and McDermott (2010) and Baur and Lucey (2010) and GARCH model |
| 2021 | CITATION | Islamic index, Bitcoin and Gold | Baur and McDermott (2010) and Baur and Lucey (2010) |

## 3. Methodology

### 3.1. Safe Haven Asset, Hedge, and Diversifier Assets

To differentiate between safe haven, hedge, and diversifier assets, we will define all three types using existing frameworks from the literature. Baur and Lucey (2010) were the first to define testable definitions of a di- versifier, hedge, and safe haven, making it possible to explore and identify the capabilities of an asset. The definitions are as follows: "A diversifier is defined as an asset that is positively (but not perfectly correlated) with another asset or portfolio on average. A hedge is an asset that is uncorrelated or negatively correlated with another asset or portfolio on average. A strict hedge is (strictly) negatively correlated with another asset or a portfolio on average. A safe haven is defined as an asset that is uncorrelated or negatively correlated with another asset or portfolio in times of market stress or turmoil."

We applied the Baur and McDermott (2010) method, which expanded on these definitions by differentiating between weak and strong forms. "A strong (weak) hedge is defined as an asset that is negatively correlated (uncorrelated) with another asset or portfolio on average. A strong (weak) safe haven is defined as an asset that is negatively correlated (uncorrelated) with another asset or portfolio in certain periods only, e.g., in times of falling stock markets.". This ability is key in the studied period, which includes the COVID pandemic.

Although the application of the method Baur and McDermott (2010) may sound plain vanilla, we have tested Multivariate GARCH and considered the DCC model and asymmetric DCC models; however, due to the extreme volatility in the cryptocurrency market, especially during the COVID pandemic, we encountered that the lack of parsimony of those methods was driving us to wrong decisions about Safe Haven Asset, Hedge and Diversifier properties, the same can be observed Hniopek and Kiselev (2015) where the reported p-values indicate poor goodness of its VAR-GARCH model. Table 1 shows that the

Baur and McDermott ([2010](#)) model continues to be used often over the years. We decided to remain with a more plain vanilla method, more appropriate to the volatile cryptocurrency market, as it gave us better control/visibility over adjusting the parameters.

### 3.2. Data Acquisition

We extracted daily historical prices, trading volumes of gold, a series of the most important cryptocurrencies (by volume for each consensus mechanism), and the most important world market indices and their correspondent money currency exchange rate from the 1st of January 2018 up to the 30th of September 2022. This data was extracted from Yahoo Finance. If more data (further into the past) was included, fewer of the newer cryptocurrencies could be included (decreasing the effectiveness of our study). Including market indexes from outside the big economies is extremely important as owners of cryptocurrencies globally come from outside big economies, according to the Statista website data Sta ([2021](#)). We also deliberately decided to include pre-COVID and during-COVID data. We analyzed the full period, pre-COVID and after-COVID started named here during-COVID.

A modeling procedure was then used to assess the hedge and safe haven properties of Bitcoin.

Table [2](#) shows descriptive statistics of the return data over the full sample period for 24 cryptocurrencies and 12 market indices, and gold.

**Table 2.** Descriptive statistics of data over the full sample period. Non US-dollar asset was converted into US-dollar using the closing exchange rate. * calculated using Close price.

| Symbol | Description | Count | Mean * | Std * | Min * | 25% * | 50% * | 75% * | Max * |
|--------|-------------|-------|--------|-------|-------|-------|-------|-------|-------|
| GC=F | Gold Spot Price | 654 | 1809.155 | 89.39434 | 1477.3 | 1751.3 | 1805.55 | 1870.075 | 2051.5 |
| BTC-USD | Bitcoin USD | 945 | 31,209.08 | 17,254.67 | 4970.788 | 12,965.89 | 32,569.85 | 45,555.99 | 67,566.83 |
| ETH-USD | Ethereum USD | 945 | 1822.112 | 1304.36 | 110.6059 | 412.4576 | 1772.102 | 2897.977 | 4812.087 |
| LTC-USD | Litecoin USD | 945 | 114.4122 | 68.22829 | 30.93088 | 53.81748 | 106.2749 | 163.8183 | 386.4508 |
| BCH-USD | BitcoinCash USD | 945 | 382.2992 | 220.6646 | 97.54175 | 233.1558 | 307.8653 | 522.2714 | 1542.425 |
| BSV-USD | BitcoinSV USD | 945 | 150.6732 | 64.37677 | 46.39974 | 94.76942 | 162.5261 | 184.2871 | 441.3943 |
| DOT-USD | Polkadot USD | 773 | 19.34745 | 12.64941 | 2.875028 | 7.365457 | 17.89226 | 28.58062 | 53.88173 |
| EOS-USD | EOS USD | 945 | 3.205723 | 1.619418 | 0.889911 | 2.358139 | 2.76872 | 3.971474 | 14.36554 |
| LINK-USD | Chainlink USD | 945 | 16.30325 | 10.13844 | 1.779877 | 7.39609 | 14.12053 | 24.63165 | 52.1987 |
| BNB-USD | BinanceCoin USD | 945 | 251.4052 | 195.5085 | 9.38605 | 28.30512 | 280.0186 | 402.45 | 675.6841 |
| VET-USD | VeChain USD | 945 | 0.055395 | 0.050509 | 0.002274 | 0.016395 | 0.031538 | 0.08584 | 0.254632 |
| ATOM-USD | Cosmos USD | 945 | 15.02725 | 11.0119 | 1.649203 | 5.281725 | 11.76493 | 23.59488 | 44.54279 |
| TRX-USD | TRON USD | 945 | 0.057196 | 0.031805 | 0.008792 | 0.026945 | 0.061292 | 0.075397 | 0.16465 |
| NEO-USD | NEO USD | 945 | 26.72435 | 20.50329 | 5.377224 | 11.06976 | 20.15465 | 38.14567 | 122.6834 |
| XTZ-USD | Tezos USD | 945 | 3.287027 | 1.492052 | 1.288307 | 2.153276 | 2.920266 | 4.062389 | 8.703033 |
| XEM-USD | NEM USD | 945 | 0.144905 | 0.120631 | 0.032336 | 0.049496 | 0.115008 | 0.185292 | 0.79527 |
| USDT-USD | Tether USD | 945 | 1.000624 | 0.002785 | 0.974248 | 1.000034 | 1.000337 | 1.000848 | 1.053585 |
| USDC-USD | USDCoin USD | 945 | 1.000184 | 0.002469 | 0.970124 | 0.999861 | 1.000072 | 1.000345 | 1.040553 |
| XMR-USD | Monero USD | 945 | 174.0921 | 82.5966 | 33.01032 | 115.8375 | 162.3905 | 230.6996 | 483.5836 |
| DASH-USD | Dash USD | 945 | 122.7475 | 72.2015 | 39.93123 | 70.98058 | 97.64275 | 164.9605 | 440.8867 |
| ZEC-USD | Zcash USD | 945 | 106.3527 | 55.31588 | 24.5043 | 61.16663 | 94.35249 | 143.9968 | 318.9179 |
| XRP-USD | Ripple USD | 945 | 0.577729 | 0.357397 | 0.139635 | 0.264122 | 0.471991 | 0.818588 | 1.839236 |
| ADA-USD | Cardano USD | 945 | 0.825132 | 0.723037 | 0.023961 | 0.129908 | 0.559813 | 1.292388 | 2.968239 |
| XLM-USD | Stellar USD | 945 | 0.215049 | 0.14156 | 0.033441 | 0.099877 | 0.185127 | 0.317452 | 0.729996 |
| CCXX-USD | CounosX USD | 895 | 56.08887 | 33.93735 | 14.45128 | 20.41959 | 53.66284 | 90.71275 | 179.0048 |

### 3.3. Data Preparation

Our final dataset was comprised of daily returns data from the 1st January 2018 to the 30th September 2020, standardized to US-dollar. Market indexes are missing when the market is closed (weekends and bank holidays), while Cryptocurrency trading has none missing as it never closes. To merge and compare the two data sets we considered (1) dropping the corresponding weekend and bank holiday from the cryptocurrency when analyzing each country index and cryptocurrency pair; (2) applying a forward filling on the missing market index data due to bank holidays and weekends (market closed). We realized that option 1, dropping Cryptocurrency data, was exaggerating our correlations.

Therefore we opted for option 2, applying a forward filling technique as it seemed a more conservative approach, and we considered that conclusions made in this scenario are valid in both scenarios.

### 3.4. Returns Calculation

Returns were calculated using daily returns:

$$R_t = \frac{P_t}{P_{t-1}}$$

where $P_t$ is the return of price $P$ at time (current day) $t$ divided by the previous day's price $P_{t-1}$.

### 3.5. Hedge or Safe Haven Assets Model

As presented in Baur and McDermott (2010), we check the safe haven, hedge, or diversifier property of all cryptocurrencies and gold against extreme conditions of different markets based on a linear regression analysis, using an ordinary least squares model. Further, we analyze the extreme conditions in the lower 10th, 5th, or 1st percentile of the return distribution as follows:

$$r_{Crypto} = c + \gamma_0 {}^* r_{M,t} + \gamma_1 {}^* D(r_M, q_{10}) {}^* r_{M,t} + \gamma_2 {}^* D(r_M, q_5) {}^* r_{M,t} + \gamma_3 {}^* D(r_M, q_1) {}^* r_{M,t} + \epsilon_t$$

where $r_{Crypto}$ is the return of the cryptocurrency estimated by the return of the market M. The constant is represented for the letter c and the error term for $\epsilon_t$. The dummy variables denoted as $D(.)$ capture extreme stock market movements and are equal to one if the stock market exceeds a certain threshold given by the 10%, 5% and 1% quantile of the return distribution.

### 3.6. Diversifier, Safe Haven or Hedge Criteria

We define the following rules, adapted from both Baur and McDermott (2010) and Shahzad et al. (2020).

neither: if one of the parameters $\gamma_1$, $\gamma_2$, and $\gamma_3$ is significantly different from zero, there is evidence of a non-linear relationship between the cryptocurrency and the stock market (regression $p$-value $\leq$ 10%).

diversi: otherwise, the cryptocurrency is considered a diversifier against movements in the market, if $\gamma_0$ is statistically significantly positive (both $\gamma_0 > 0.0$ and regression $p$-value $> 10\%$)

haven+: otherwise, the cryptocurrency is a strong safe haven if the quantile coefficients $\gamma_1$, $\gamma_2$, and $\gamma_3$ are all statistically negative ($\gamma_{1,2,3} < 0.0$ and regression $p$-values $\leq 10\%$)

haven-: otherwise, it is a weak safe haven if the quantile coefficients $\gamma_1$, $\gamma_2$, and $\gamma_3$ are negative or statistically insignificantly different from zero ($\gamma_{1,2,3} < 0.0$ or regression $p$-values $> 10\%$)

hedge+: otherwise, it is a strong hedge if $\gamma_0$ is statistically significantly negative (both $\gamma_0 < 0.0$ and regression $p$-value $\leq 10\%$)

hedge-: otherwise, it is a weak hedge if $\gamma_0$ is zero (regression $p$-value $> 10\%$);

## 4. Results and Discussion

Table 3 presents some qualitative information about each asset (gold and cryptocurrency) used in the analysis; we can observe the year the cryptocurrency was first released to the market; some special cases have the month also indicated as our data selection starts in 1st of January 2018 because we only have partial data for this cryptocurrency, however, for some interesting conclusion can be withdrawn even with partial data. The definition of all Blockchain Technologies (Wik 2021a, 2021b, 2021c) (PoW, PoS, PoA, SCP, dBFT, PoI, Stablecoin, and a mix of PoW & PoS) can be found in Appendix A. Tables 3–6 are sorted in

descending order by volume of transactions at the 30th of September 2022. This acts as an interesting proxy for market adoption.

**Table 3.** Qualitative information for each asset (gold and cryptocurrency) used in the analysis.

| Symbol | Description | Type | Technology | Release | Volume Tril. $ |
|---|---|---|---|---|---|
| GC=F | Gold Spot Price | gold | - | - | 548 |
| USDT-USD | Tether USD | crypto-public | Stablecoin | 2015 | 27.31292 |
| BTC-USD | Bitcoin USD | crypto-public | PoW | 2009 | 18.71954 |
| ETH-USD | Ethereum USD | crypto-public | PoW | 2015 | 6.227961 |
| USDC-USD | USDCoin USD | crypto-public | Stablecoin | September/2018 | 3.108134 |
| XRP-USD | Ripple USD | crypto-public | Consensus | 2013 | 1.544058 |
| BNB-USD | BinanceCoin USD | crypto-public | PoS | 2017 | 0.593376 |
| ADA-USD | Cardano USD | crypto-public | PoS | 2017 | 0.317806 |
| LINK-USD | Chainlink USD | crypto-public | PoS | May/2019 | 0.313342 |
| TRX-USD | TRON USD | crypto-public | PoS | 2017 | 0.292353 |
| XLM-USD | Stellar USD | crypto-public | SCP | 2014 | 0.267535 |
| LTC-USD | Litecoin USD | crypto-public | PoW | 2011 | 0.256914 |
| BCH-USD | BitcoinCash USD | crypto-public | PoW | 2017 | 0.202425 |
| ATOM-USD | Cosmos USD | crypto-public | PoS | March/2019 | 0.17939 |
| DOT-USD | Polkadot USD | crypto-public | PoS | May/2020 | 0.177865 |
| EOS-USD | EOS USD | crypto-public | PoS | 2017 | 0.163115 |
| XMR-USD | Monero USD | crypto-private | PoW | 2014 | 0.060837 |
| BSV-USD | BitcoinSV USD | crypto-public | PoW | May/2018 | 0.052668 |
| DASH-USD | Dash USD | crypto-private | PoW | 2014 | 0.050602 |
| ZEC-USD | Zcash USD | crypto-private | PoW | 2016 | 0.047063 |
| VET-USD | VeChain USD | crypto-public | PoA | 2016 | 0.045363 |
| NEO-USD | NEO USD | crypto-public | dBFT | 2014 | 0.023583 |
| XTZ-USD | Tezos USD | crypto-public | PoS | 2017 | 0.013301 |
| XEM-USD | NEM USD | crypto-public | PoI | 2015 | 0.007806 |
| CCXX-USD | CounosX USD | crypto-public | mix PoW & PoS | June/2019 | 0.000578 |

Tables 4–6 present the results obtained for gold and all analyzed cryptocurrency for the BRICS and G7 market indexes during the whole period analyzed, ranging from the 1st of January 2018 to the 30th of September 2022, the pre-COVID period ranging from the 1st of January 2018 to the 29th of February 2020 and during-COVID, the period after COVID health crisis started ranging from the 1st of March 2020 to the 30th of September 2022, respectively.

From the whole period in Table 3, we observed the following:

- Stablecoins Tether (USDT) and USDCoin (USDC) are the most effective hedge/havens compared to other cryptocurrencies.
- Gold acts as a diversifier asset for the majority of indices. Gold can be used for hedging in India and Italy. However, it had no interesting property for Russia and the UK.
- China, Russia, Italy, and the UK appear to be outliers, with a majority of "neither" for most cryptocurrencies.
- In South Africa, Canada, France, and Germany, most cryptocurrencies.
- From the Pre-COVID period Table 4, we observed the following:
- In some markets, Gold is used to hedge against the other assets' risk.
- Investors in China use multiple Cryptocurrencies as hedge/haven against risk in potentially bad scenarios for their economy.
- Most cryptocurrency acts as a hedge/haven for France, Italy, and Germany, indicating the use of cryptocurrency in European nations to mitigate risk.
- From the during-COVID period Table 5, we observed the following:
- Gold lost hedge properties in most markets and is used to diversify from other assets, but not with offsetting positions (to hedge against the other assets' risk).
- The Chinese market stopped being correlated with any cryptocurrency. This is likely related to China's strong regulations on cryptocurrencies, especially in early 2021.
- Most countries continued using Tethet (USTD) and USDCoin (USDC) as hedging methods.
- In France, we can observe a shift from hedging to diversifying, meaning a loss of hedging property in those markets.

**Table 4.** Whole Period—Baur and McDermott Model for the BRICS and G7 countries (daily returns in US-dollar for the whole period from 1st of January 2018 to 30th of September 2022), "-" means neither hedge, nor haven, nor diversifier. diversifier: moves in a the same direction of the market index, hedging: moves in the opposite direction of the market index, haven: or safe haven, moves in the opposite direction of the market index in extreme cases. -/+ means often/most cases.

| ^BVSP | 000001.SS | ^BSESN | IMOEX.ME | ^JN0U.JO | ^GSPTSE | ^FCHI | ^GDAXI | ^ITLMS.MI | ^N225 | ^FTSE | ^GSPC |
|---|---|---|---|---|---|---|---|---|---|---|---|
| IBOVESPA | SSE | SENSEX | MOEX | TRI | TSX | CAC 40 | DAX | FTSE Italia | Nikkei 225 | FTSE 100 | S&P 500 |
| Brazil | China | India | Russia | South Africa | Canada | France | Germany | Italy | Japan | UK | USA |
| BRICS | BRICS | BRICS | BRICS | BRICS | G7 | G7 | G7 | G7 | G7 | G7 | G7 |
| diversi | diversi | hedge- | - | diversi | diversi | diversi | diversi | hedge+ | diversi | - | diversi |
| hedge+ | hedge- | hedge+ | hedge- | hedge+ | hedge+ | hedge+ | hedge+ | hedge- | diversi | - | hedge+ |
| diversi | - | diversi | hedge- | diversi | diversi | diversi | diversi | diversi | - | - | diversi |
| - | - | diversi | hedge- | diversi | diversi | diversi | diversi | - | - | - | hedge- |
| hedge+ | haven- | hedge+ | hedge- | hedge+ | hedge+ | hedge+ | hedge+ | hedge- | diversi | - | hedge+ |
| - | hedge- | hedge- | - | - | - | diversi | diversi | - | - | - | - |
| - | - | diversi | - | diversi | diversi | diversi | diversi | - | diversi | - | hedge- |
| - | - | hedge- | - | hedge- | - | - | - | - | - | - | hedge- |
| - | - | - | - | - | diversi | diversi | diversi | - | - | - | - |
| - | hedge- | - | - | diversi | diversi | diversi | diversi | - | - | - | - |
| - | - | hedge- | - | - | - | - | - | - | - | hedge- | - |
| diversi | - | diversi | - | diversi | diversi | diversi | diversi | - | diversi | - | diversi |
| diversi | - | - | - | diversi | diversi | diversi | diversi | - | - | - | diversi |
| haven- | - | - | - | - | diversi | - | diversi | diversi | - | - | - |
| diversi | - | - | - | hedge- | - | diversi | - | - | diversi | - | - |
| diversi | - | - | - | diversi | diversi | diversi | diversi | - | - | - | diversi |
| haven- | - | diversi | - | diversi | diversi | diversi | diversi | - | diversi | - | diversi |
| diversi | - | diversi | haven- | diversi | diversi | diversi | diversi | - | - | - | diversi |
| diversi | - | - | - | diversi | diversi | diversi | diversi | - | - | - | - |
| haven- | - | - | - | diversi | diversi | diversi | diversi | - | - | - | - |
| hedge- | - | - | - | - | - | - | - | - | diversi | - | - |
| - | - | - | - | - | - | - | - | - | hedge- | - | - |
| diversi | - | diversi | hedge- | diversi | diversi | diversi | diversi | - | - | - | diversi |
| - | - | - | - | - | - | - | - | - | hedge- | - | - |
| - | - | - | - | - | - | - | - | - | diversi | - | - |

**Table 5.** Pre-COVID—Baur and McDermott Model for the BRICS and G7 countries (daily returns in US-dollar for the PRE-COVID from 1st of January 2018 to 29th of February 2020), "-" means neither hedge, nor haven, nor diversifier. diversifier: moves in a the same direction of the market index, hedging: moves in the opposite direction of the market index, haven: or safe haven, moves in the opposite direction of the market index in extreme cases. -/+ means often/most cases.

| Description | ^BVSP | 000001.SS | ^BSESN | IMOEX.ME | ^JN0U.JO | ^GSPTSE | ^FCHI | ^GDAXI | ^ITLMS.MI | ^N225 | ^FTSE | ^GSPC |
|---|---|---|---|---|---|---|---|---|---|---|---|---|
| | IBOVESPA | SSE | SENSEX | MOEX | TRI | TSX | CAC 40 | DAX | FTSE Italia | Nikkei 225 | FTSE 100 | S&P 500 |
| | Brazil | China | India | Russia | South Africa | Canada | France | Germany | Italy | Japan | UK | USA |
| | BRICS | BRICS | BRICS | BRICS | BRICS | G7 | G7 | G7 | G7 | G7 | G7 | G7 |
| Gold Spot Price | - | hedge- | diversi | diversi | diversi | diversi | hedge+ | - | hedge- | hedge- | hedge- | - |
| Tether USD | - | - | - | - | - | - | hedge- | haven- | - | haven- | - | - |
| Bitcoin USD | - | hedge+ | haven- | - | - | diversi | hedge+ | haven- | hedge- | haven- | - | - |
| Ethereum USD | diversi | haven- | hedge- | - | - | diversi | hedge- | hedge- | hedge- | hedge- | - | diversi |
| USDCoin USD | diversi | haven- | - | - | - | - | hedge- | hedge- | hedge- | - | - | - |
| Ripple USD | haven- - | - | - | - | hedge- | diversi | hedge- | haven- | - | hedge- | - | - |
| BinanceCoin USD | - | haven- | - | - | - | diversi | - | - | hedge- | hedge- | - | - |
| Cardano USD | diversi | - | - | haven- | - | - | - | - | - | - | - | - |
| Chainlink USD | - | - | - | - | - | diversi | - | - | hedge- | haven- | - | - |
| TRON USD | - | - | - | - | - | - | - | - | hedge- | - | - | - |
| Stellar USD | haven- | haven- | - | diversi | hedge- | - | - | - | - | haven- | hedge+ | - |
| Litecoin USD | diversi | haven- | - | - | - | diversi | hedge- | hedge+ | hedge- | hedge- | - | - |
| BitcoinCash USD | haven- | - | - | - | - | diversi | hedge- | hedge- | - | hedge+ | - | diversi |
| Cosmos USD | diversi | haven- | - | - | - | - | - | hedge- | - | - | - | haven- |
| Polkadot USD | diversi | - | hedge- | - | haven- | haven- | - | - | - | - | - | - |
| EOS USD | - | - | - | - | - | - | - | - | hedge- | haven- | - | - |
| Monero USD | diversi | haven- | - | - | - | diversi | hedge- | hedge+ | hedge- | hedge+ | - | - |
| BitcoinSV USD | - | hedge- | - | - | diversi | - | hedge- | diversi | hedge- | diversi | - | - |
| Dash USD | diversi | haven- | - | - | - | - | - | - | - | - | - | - |
| Zcash USD | diversi | haven- | - | - | hedge- | - | - | - | hedge- | - | - | - |
| VeChain USD | haven- | haven- | - | - | hedge- | diversi | hedge- | haven- | haven- | hedge- | - | - |
| NEO USD | - | - | - | - | - | - | - | - | - | hedge- | - | - |
| Tezos USD | - | - | - | hedge- | - | - | hedge- | hedge- | - | hedge+ | - | - |
| NEM USD | neither | neither | neither | neither | neither | neither | neither | neither | neithe | hedge- | neither | neither |
| CounosX USD | neither | neither | neither | neither | neither | neither | neither | neither | neithe | diversi | neither | neither |

**Table 6.** During-COVID—Baur and McDermott Model for the BRICS and G7 countries (daily returns in US-dollar for the DURING-COVID period from 1st of March 2020 to 30th of September 2020), "-" means neither hedge, nor haven, nor diversifier. diversifier: moves in a the same direction of the market index, hedging: moves in the opposite direction of the market index, haven: or safe haven, moves in the opposite direction of the market index in extreme cases. -/+ means often/most cases.

| Description | ^BVSP | 000001.SS | ^BSESN | IMOEX.ME | ^JN0U.JO | ^GSPTSE | ^FCHI | ^GDAXI | ^ITLMS.MI | ^N225 | ^FTSE | ^GSPC |
|---|---|---|---|---|---|---|---|---|---|---|---|---|
| | IBOVESPA | SSE | SENSEX | MOEX | TRI | TSX | CAC 40 | DAX | FTSE Italia | Nikkei 225 | FTSE 100 | S&P 500 |
| | Brazil | China | India | Russia | South Africa | Canada | France | Germany | Italy | Japan | UK | USA |
| | BRICS | BRICS | BRICS | BRICS | BRICS | G7 | G7 | G7 | G7 | G7 | G7 | G7 |
| Gold Spot Price | diversi | diversi | hedge- | - | diversi | diversi | diversi | diversi | hedge+ | diversi | - | diversi |
| Tether USD | hedge+ | - | hedge+ | hedge- | hedge+ | hedge+ | hedge+ | hedge+ | hedge- | diversi | haven- | hedge+ |
| Bitcoin USD | diversi | - | diversi | - | diversi | diversi | diversi | diversi | diversi | - | - | diversi |
| Ethereum USD | hedge- | - | diversi | - | diversi | diversi | diversi | diversi | - | - | - | hedge- |
| USDCoin USD | hedge+ | - | hedge+ | hedge- | hedge+ | hedge+ | hedge+ | hedge+ | hedge- | diversi | haven- | hedge+ |
| Ripple USD | haven- | - | hedge- | - | diversi | - | diversi | diversi | - | - | - | - |
| BinanceCoin USD | hedge- | - | hedge- | - | diversi | - | diversi | diversi | - | diversi | - | hedge- |
| Cardano USD | hedge- | - | hedge- | - | hedge- | - | - | - | - | - | - | - |
| Chainlink USD | haven- | - | - | - | - | - | diversi | diversi | - | - | - | - |
| TRON USD | - | - | diversi | - | diversi | diversi | diversi | diversi | - | - | - | - |
| Stellar USD | - | - | hedge- | - | - | - | - | - | - | - | hedge- | - |
| Litecoin USD | diversi | - | - | - | diversi | diversi | diversi | diversi | - | - | - | diversi |
| BitcoinCash USD | diversi | - | - | - | diversi | diversi | diversi | diversi | - | hedge- | - | diversi |
| Cosmos USD | - | - | - | - | - | - | diversi | - | diversi | - | - | - |
| Polkadot USD | diversi | - | - | - | hedge- | - | diversi | - | - | diversi | - | - |
| EOS USD | diversi | - | - | - | diversi | diversi | diversi | diversi | - | hedge- | - | diversi |
| Monero USD | hedge- | - | diversi | - | diversi | diversi | diversi | diversi | - | - | - | hedge- |
| BitcoinSV USD | diversi | - | diversi | hedge- | diversi | diversi | diversi | diversi | - | hedge- | - | diversi |
| Dash USD | diversi | - | - | - | diversi | diversi | diversi | diversi | - | hedge- | - | diversi |
| Zcash USD | - | - | - | - | diversi | diversi | diversi | diversi | - | - | - | - |
| VeChain USD | - | - | - | - | - | - | - | - | - | - | - | - |
| NEO USD | - | - | - | - | - | - | - | diversi | - | diversi | - | - |
| Tezos USD | diversi | - | diversi | - | diversi | diversi | diversi | diversi | - | hedge- | - | diversi |
| NEM USD | - | - | - | - | - | - | - | - | - | - | - | - |
| CounosX USD | - | - | - | - | - | - | - | - | - | diversi | - | - |

## 5. Conclusions and Outlook

In this paper, we report the results of studying the repeating patterns by Blockchain Technology and maturity (market adoption) of Hedge, Safe Haven, and Diversifier properties of a very extensive list of Cryptocurrencies (Cardano, Cosmos, Cosmos, BitcoinCash, BinanceCoin, BitcoinSV, Bitcoin, CounosX, Dash, Polkadot, Polkadot, EOS, Ethereum, Chainlink, Litecoin, NEO, TRON, USDCoin, Tether, VeChain, NEM, Stellar, Monero, XRP, Tezos, Zcash) with daily returns in the period of 1st of January 2018 up to the 30th of September 2022, first by the G7 (Canada, France, Germany, Italy, Japan, UK, and the USA) major market indexes as well as all BRICS (Brazil, China, India, Russia, and South Africa) major market indexes. The period was specially chosen in order to contain data for a great number of the newer cryptocurrencies.

Our results show that gold lost the property of hedging instruments from the pre-COVID to the During-COVID period. We demonstrated that Stablecoins Tether (USDT) and USDCoin (USDC), on the other hand, are effective hedging instruments as the hedging property remained that way for most markets, including S&P 500.

Our results support Bouri et al.'s (2017) conclusions that Bitcoin is an effective diversifier in the During-COVID period and that Bitcoin safe haven property can be found in Japan and China. However, our result expands that demonstrating a shift in properties from hedging to diversifier in Bitcoin since the outbreak of the COVID crisis. Bitcoin, Ethereum, Litecoin, and BitcoinCash behave similarly in all markets, mainly as diversifiers, indicating that investors use Pulic-PoW cryptocurrencies with the same strategy interchangeably. The exception is the immature BitcoinSV.

Our results show that the Chinese market stopped being correlated with any cryptocurrency after the COVID health crisis started; this is likely to be related to China's strong regulations on cryptocurrencies, especially in early 2021. The same happens to the Russian market, but more data is necessary to see if this behavior has to do with COVID or the war in Ukraine.

This paper and its results are not alone as it is related to a recent strand of literature concerning the hedging and safe haven properties of Bitcoin in general and other cryptocurrencies, including stablecoins, in particular, and the effect of uncertainty periods such as the pandemic, as in the following studies: Usman and Nduka (2022); Bouri et al. (2018) and Shahzad et al. (2022).

These papers find evidence that the relationship between cryptocurrencies and uncertainty measures is not the same across various quantiles. In this regard, it supports our main conclusion of the special role of the underlying technology (consensus mechanism) of the cryptocurrency as a possible explanation for its market adoption and its relationship with cryptocurrencies, and the uncertainty measures observed.

## 6. Future Research

China initiated back in 2014 its Digital Yuan project designed to replace the cash and coin in circulation; real-world trials of the Digital Yuan are already underway in 2021 Kharpal (2021). It is important to disclaim that it is not a Cryptocurrency but a Central Bank-backed digital currency, so further investigation is needed to assess how Digital Yuan will affect the Cryptocurrency market and if any volatility spillovers can be observed.

An element to consider for analyzing the formation of the price and economic structure (Safe Have, Hedge, Diversifier) of Cryptocurrencies is changes in the policy and political landscape, specifically those changes captured in the economic policy uncertainty (EPU) index. Although it can be said that this also affects gold and other assets, further investigation is necessary to assess how Cryptocurrencies react to changes in the EPU index Demir et al. (2018).

For future research on this topic, another interesting aspect requiring further investigation is a study of the path to maturity of a cryptocurrency (volume and/or time required for maturity). For example, BitcoinSV USD is a low-volume newly released public PoW

cryptocurrency that is still not behaving as the rest of the public PoW cryptocurrencies analyzed in this study (Bitcoin USD, Ethereum USD, Litecoin USD, BitcoinCash USD).

It would be interesting to delve into Private Cryptocurrencies (Monero, Dash, and Zcash) in order to investigate further the uses of these assets as they seem unrelated to the rest.

Also, a more detailed discussion of country policy implications during the studied period would be very informative for crypto-traders, investors, and policymakers. Further, it would be interesting to repeat the study with a breakdown of the analyzed period by pre-COVID and during-COVID crisis, potentially applying methods like Detrended Fluctuation Analysis (DFA) and the Detrended Cross-Correlation Analysis used in the paper Ferreira (2018) and also wavelet value-at-risk used in the paper Bouri et al. (2020) when studying in the future the relationship across major cryptocurrencies and attractiveness measures since market participants in the cryptocurrency markets have various investment horizons.

Finally, the current work can be improved with a robust statistical comparison of methodologies used for Safe Haven properties, including GARCH, DCC-GARCH, Smooth transition regression (STR) model, Quantile correlation approach, and Detrended partial cross-correlation analysis (DPCCA) and use both diagnostic tests (R-squared and other metrics) and linearity and multicollinearity assumption tests to assess the most appropriate technique for the problem.

**Author Contributions:** Conceptualization, M.-A.S. and M.F.A.G.; methodology, M.-A.S.; software, M.-A.S.; validation, M.-A.S. and M.F.A.G.; formal analysis, M.F.A.G.; investigation, M.F.A.G.; resources, M.F.A.G.; data curation, M.F.A.G.; writing—original draft preparation, M.-A.S. and M.F.A.G.; writing—review and editing, M.-A.S. and M.F.A.G.; visualization, M.-A.S. and M.F.A.G.; supervision, M.F.A.G.; project administration, M.F.A.G.; funding acquisition, no funding was used. All authors have read and agreed to the published version of the manuscript.

**Funding:** This research received no external funding.

**Data Availability Statement:** For reproducibility purposes, all datasets and Jupyter Noteboks of the project are available under the DOI Badge 10.5281/zenodo.7380163 (https://zenodo.org/badge/latestdoi/559297740 (accessed on 1 November 2022)) and also under the project's Github repository: https://github.com/manoelgadi/CryptoSafeHavenHedgeAndDiversification (accessed on 1 November 2022).

**Conflicts of Interest:** The authors declare no conflict of interest.

## Appendix A. Blockchain Technology Appendix

Texts extracted from referenced sites.

- PoW: Proof of work (PoW) is a form of cryptographic zero-knowledge proof in which one party (the prover) proves to others (the verifiers) that a certain amount of a specific computational effort has been expended. Verifiers can subsequently confirm this expenditure with minimal effort on their part Wik (2021a).
- PoS: Proof of stake (PoS) protocols are a class of consensus mechanisms for blockchains that work by selecting validators in proportion to their quantity of holdings in the associated cryptocurrency. Unlike a proof of work (PoW) protocol, PoS systems do not incentivize extreme energy consumption. The first functioning use of PoS for cryptocurrency was Peercoin in 2012. Other uses have followed, and the Ethereum Foundation has announced a plan to switch Ethereum from PoW to PoS within 2021 Wik (2021b).
- PoA: Proof of authority (PoA) is an algorithm with blockchains that delivers comparatively fast transactions through a consensus mechanism based on identity as a stake. The most notable platform using PoA is VeChain Wik (2021c).
- SCP: Stellar Consensus Protocol (SCP). The Stellar Consensus Protocol was first described in a whitepaper by David Mazi'eres in 2015. It is a "federated Byzantine

agreement system" that allows decentralized, leaderless computing networks efficiently to reach a consensus outcome on some decisions (Mazieres 2015).

- PoI: Proof of importance (POI) and harvesting. NEM addresses the issue using its POI mechanism, as it gives more "importance" to how much one is "invested" into the NEM system, with realistic "vested" interest. The XEM coins in the wallet and the holding period play a key role in gauging the importance (2021).
- dBFT: Delegated Byzantine Fault Tolerance (dBFT 2021) is a sophisticated algorithm meant to facilitate consensus on a blockchain. Although it is not in common use as of yet, it represents an alternative to simpler proof of stake, proof of importance and proof of work methods.
- Stablecoin: Stablecoins are cryptocurrencies where the price is designed to be pegged to a cryptocurrency, fiat money, or to exchange-traded commodities Stablecoin Wik (2021a). USD Coin (USDC) is a type of cryptocurrency that is referred to as a stablecoin; one can always redeem 1 USD Coin for US$1.00, giving it a stable price Stablecoin Wik (2021b).

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
