# Peer review of "Analyzing Safe Haven, Hedging and Diversifier Characteristics of Heterogeneous Cryptocurrencies against G7 and BRICS Market Indexes"

_jrfm, doi:10.3390/jrfm15120572_

Round 1

Reviewer 1 Report

 Review Report: jrfm-1990542

I have doubts about the novelty of this work is enough for a top journal as Journal of Risk and Financial Management. Then, with regret, I recommend its rejection.

In order to improve the presentation of the manuscript, maybe the authors want to consider the following comments: 

1-     The abstract is long and contains many abbreviations without definition, the authors must remove more details, and only put a good summary of the paper.

2-     Define each abbreviation that is not defined in the paper.

3-     The authors should provide the statistical details of the model in subsection 3.5, such as the assumptions, estimation method that used, and so on.

4-     In Data Preparation section, authors must test the outliers in the data before estimating the model. Furthermore, authors must test the multicollinearity between the independent variables.

5-     The authors should perform the diagnostic tests after the estimation. 

6-     In general, the statistical analysis of the paper needs more depth.

 ================================================

Author Response

Dear reviewer,

We hope that the point-by-point responses given in the PDF attached, our commitment for making the corrections and adaptations requested are sufficient to convivence you of our commitment with this research and help you accepting the paper for publication.

Thanks,
The authors.

Reviewer 2 Report

Article can be published without changes.

Author Response

Dear reviewer, 

Thank for acepting the paper for publication without changes.

Best Regards,
The authors.

Reviewer 3 Report

The Researcher carried out the research study in a predominant way with clear justification and provides more insights about the intended study to targeted audience. Overall Good Research Work

Author Response

(The authors gave the same response as above.)

Reviewer 4 Report

The contribution of the paper is low since the methods employed are well known and results are not interesting from a practical standpoint. Literal presentation is poor, written style has room for improvement. Several recent approaches were omitted in literature review. Performance evaluation of the methods do not include significant figures of merit for the proposed application. The discussions have not been done in depth.

-       A rationale of the selection of the methods and tuning of all the parameters an explanation of the selected features (from the user standpoint) should be added.

-       The introductory part of this paper did not systematically address the problems of existing methods, and the highlights of this work were not highlighted in the paper.  It is recommended that the authors further summarize the contributions of this work from the common problems of existing methods.

- While discussing results, it is interesting to justify the obtained results by mentioning advantages and disadvantages of the compared techniques, which is absent in this paper.

- The motivation is not clear. Please specify the importance of the proposed solution

-More mathematical analysis and related equations should be given.

- The state of the art is very limited; although they claimed that the subject is of great interest? They should present a categorical view of different existing approaches. In the proposed technique; the authors consumed a large space in primitives related to the basic ideas.

Author Response

Dear reviewer, 

Hope the point-by-point responses provided in the PDF attached demonstrate our commitment to improve our research with your valuable input and our commitment with JRFM and is sufficient to convince you to accept the paper for publication.

Thanks!

Best Regards,
The authors.

Round 2

Reviewer 1 Report

I have checked the revised version carefully. I am happy with the corrections in the revised manuscript. It was improved. So, the current version of this manuscript is suitable for publication. 

Good Luck.